# Experimental glycopeptide antibiotic EVG7 prevents recurrent *Clostridioides difficile* infection by sparing members of the *Lachnospiraceae* family

Elma Mons [1], Jannie G. E. Henderickx [2], Ingrid M. J. G. Sanders[3], Anusca G. Rader [3], Caroline E. Perkins[4], Florence M. Stel [1], Emma van Groesen [1], Wiep Klaas Smits [2,3], Casey M. Theriot [4] & Nathaniel I. Martin [1] ✉

Oral vancomycin has a long history as the first-line treatment for *Clostridioides difficile* infection (CDI), but its use is associated with high relapse rates. Antibiotics that more selectively target *C. difficile* while sparing protective commensal gut bacteria, have the potential to prevent recurrent CDI (rCDI). Here, we investigate the experimental glycopeptide antibiotic, EVG7, in the context of rCDI. In vitro susceptibility assays reveal that clinical *C. difficile* isolates are up to 16-times more sensitive to EVG7 (MIC = 0.063–0.25 mg/L) compared to vancomycin (MIC = 0.5–2 mg/L). In a validated mouse model of rCDI in male mice, low dose oral EVG7 (0.04 mg/mL in drinking water) more effectively treats primary CDI and prevents recurrence, outperforming a 10-fold higher dose of vancomycin. Subsequent microbiome analysis and in vitro susceptibility testing reveal that EVG7 preserves *Lachnospiraceae*, a family of commensal bacteria associated with protection against *C. difficile* colonization.

*Clostridioides difficile* is a Gram-positive anaerobic pathogen that is the leading cause of healthcare- and antibiotic-associated diarrhea in the United States and Europe[1–3]. Clinical symptoms of *C. difficile* infection (CDI) range from mild diarrhea to life-threatening colitis and death[4]. Each year an estimated 0.5 million patients are affected by CDI in the U.S., of which 30,000 infections are lethal[1]. CDI also contributes to a significant demand on healthcare resources with associated costs between $71,980 (no recurrence) and $207,733 (≥3 recurrences) per patient, amounting to an annual economic burden of around $5.4 billion[5]. Aside from recent antibiotic exposure, patient-specific risk factors for severe CDI are hospitalization and advanced age. Infection with *C. difficile* typically happens by transmission of heat-, acid-, oxygen- and antibiotic-resistant spores[4]. These spores are not killed by routine cleaning with alcohol-based sanitizers and can survive on surfaces for several months[4], which is of particular concern in hospitals and care facilities where such spores are ubiquitously present[6]. Intestinal colonization can be asymptomatic, but active CDI will develop when the dormant spores germinate into anaerobic, toxin-producing vegetative cells in the presence of host-derived bile salt germinants and amino acids[7]. Antibiotics remain a major risk factor for CDI as they disrupt the composition and function of the gut microbiome, thus generating a favorable environment for *C. difficile* germination, outgrowth, and toxin production[8].

Despite contributing to the onset of CDI, antibiotics remain the standard of care treatment for CDI, typically as a 10-day course of oral vancomycin or fidaxomicin[9–11]. Oral vancomycin – and fidaxomicin to a lesser extent – is associated with CDI recurrence, with up to 25% of

[1]Biological Chemistry Group, Institute of Biology Leiden, Leiden University, Leiden, Netherlands. [2]Center for Microbiome Analyses and Therapeutics, Leiden University Center for Infectious Diseases (LUCID), Leiden University Medical Center, Leiden, Netherlands. [3]Experimental Bacteriology Group, Leiden University Center for Infectious Diseases (LUCID), Leiden University Medical Center, Leiden, Netherlands. [4]Department of Population Health and Pathobiology, College of Veterinary Medicine, NC State University, Raleigh, NC, USA. ✉e-mail: n.i.martin@biology.leidenuniv.nl

patients relapsing within 2–8 weeks after antibiotic treatment for the initial episode[12]. Probability of recurrence increases with each episode[12] and can be attributed to disruption of the indigenous gut microbiota by these antibiotics, resulting in an altered environment that allows persistent *C. difficile* spores to regerminate[7,13]. While historically metronidazole was also used for treating CDI, increasingly poor cure rates attributed to reduced susceptibility have led to it being deprioritized in the clinical guidelines[10,14,15]. For fidaxomicin, reports have revealed emerging resistance in *C. difficile*[16,17]. Given that fidaxomicin only recently made it into the clinical guidelines[10,15], such reports of resistance may be a cause for concern[18]. With regard to vancomycin, cases of reduced susceptibility leading to poorer clinical outcomes have also been reported[19], however resistance to vancomycin is still uncommon in *C. difficile* despite a long history of use. There are also concerns that oral vancomycin can select for resistance in other gut bacteria, for example leading to further colonization with pathogenic vancomycin-resistant *enterococci* (VRE) or *S. aureus* (VRSA)[20]. As an alternative to traditional antibiotics for treatment of recurrent CDI (rCDI), fecal microbiota transplantation (FMT) and more recently microbiota-focused therapies and live biotherapeutic products (LBPs) have been developed[21–24]. However, such treatments are adjuvant at best and do not replace antibiotic therapy[25,26]. Furthermore, reliable access to FMT products can be challenging[27,28] and the two FDA-approved microbiota-focused therapies, VOWST[29] and REBYOTA[30], are not approved for treatment of pediatric patients[31] or patients with fulminant CDI[32].

These issues underscore the need for new antibiotics that target *C. difficile* while sparing other members of the gut microbiota, thus preventing recurrence. Recently, we reported the discovery and development of the new glycopeptide antibiotic EVG7 that holds promise as a treatment for challenging Gram-positive infections[33]. Currently in preclinical development, EVG7 is a semisynthetic vancomycin derivative with an improved safety profile that outperformed vancomycin against various (drug-resistant) Gram-positive pathogens, including vancomycin-resistant strains (e.g., VRE, VRSA), with low propensity for resistance selection. EVG7 was also found to be safe and well-tolerated in murine models.

In this work, we show that EVG7 also outperforms vancomycin against a panel of different human *C. difficile* clinical isolates. We further show that in a validated mouse model of rCDI a lower dose of oral EVG7 effectively treats primary CDI while preventing recurrence, demonstrating superiority to standard of care vancomycin. Notably, the reduced recurrence of CDI in EVG7-treated mice is paralleled by the preservation of *Lachnospiraceae*, a family of commensal bacteria associated with protection against *C. difficile* colonization in the gut.

## Results

### EVG7 outperforms vancomycin against *C. difficile* clinical isolates

To evaluate whether EVG7 is more potent than vancomycin (VAN) against *C. difficile*, in vitro susceptibility was tested against a panel of clinically relevant *C. difficile* isolates belonging to common phylogenic clades (Fig. 1, Supplementary Table 1). Agar dilution susceptibility testing of this collection revealed $MIC_{50}$-values of 1 µg/mL for vancomycin and 0.125 µg/mL for EVG7, corresponding to an 8-fold higher sensitivity to EVG7. Across all phylogenic clades tested, the *C. difficile* MICs for EVG7 were consistently 8-fold to 16-fold lower than for vancomycin. Notably, susceptibility was similar in clades associated with epidemics and outbreaks[18], such as clade 2 – which includes PCR ribotypes (RT) 027 and 176/multilocus sequence type (ST)1 – and clade 5 which includes RT078/ST11.

### Oral EVG7 prevents recurrence of CDI in vivo while vancomycin does not

Having established its in vitro activity against *C. difficile*, EVG7 was further evaluated in a mouse model of rCDI (Fig. 2a). In this established

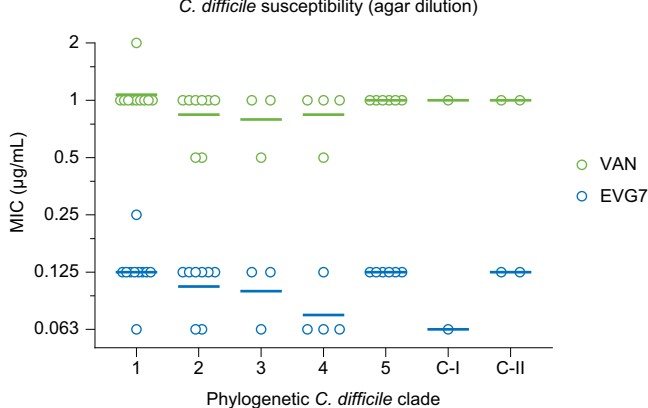

**Fig. 1 | EVG7 exhibits potent in vitro activity against human *C. difficile* isolates.** Minimal Inhibitory Concentration (MIC) of vancomycin (VAN) and EVG7 against *C. difficile* clinical isolates, determined by agar dilution method in Brucella Blood Agar (BBA) supplemented with sheep blood, hemin, and vitamin K. The isolates tested belong to a well-established reference collection[88,89] supplemented with clinical isolates collected by the National Expertise Center for *C. difficile* infections (hosted at the Leiden University Medical Center (LUMC), the Netherlands)[106]. Each dot represents the MIC value for an independent single isolate, the lines represent the geometric mean MIC value in this clade for each compound (independent biological replicates). Graphs were generated in GraphPad Prism 10 and further compiled in Adobe Illustrator 2024. Source data are provided in Supplementary Table 1 and as a Source Data file.

model[34–36], male mice were pretreated with the broad-spectrum antibiotic cefoperazone for five days, followed by two days on regular water, to render them susceptible to colonization. Mice were challenged with *C. difficile* spores on day 0 leading to established primary CDI by day 4. Next, drinking water supplemented with antibiotic (vancomycin or EVG7) was made available ad libitum for 5 days, after which antibiotic treatment was ceased and animals were monitored for clinical signs of rCDI. Clinical signs of disease were measured by monitoring weight loss (Fig. 2b) and utilizing a clinical scoring system to categorize severity of CDI (Supplementary Table 2, Fig. 2c). In this study, mice were divided into three groups: one 'no CDI' group (n = 4) that was not challenged with *C. difficile* and two antibiotic groups that were challenged with *C. difficile*: the comparator 0.4 mg/mL vancomycin group (VAN, n = 8), that typically relapses after 13–15 days, and the 0.04 mg/mL EVG7 group (n = 12). The 10-fold lower dose of oral EVG7 was selected in accordance with the higher in vitro sensitivity of *C. difficile* isolates to EVG7.

As expected, both antibiotic-treated groups exhibited weight loss (Fig. 2b) and clinical signs of disease (Fig. 2c) in the first four days following *C. difficile* spore challenge, signifying primary CDI. However, mice receiving 0.4 mg/mL vancomycin in their drinking water slowly started to show clinical signs of relapse after antibiotic treatment was halted on day 9, including significant weight loss compared to both the 'no CDI' and EVG7 groups (P ≤ 0.0001), and had to be euthanized on day 14/15. While the total *C. difficile* load (vegetative cells and spores) in the feces was lowered almost to the limit of detection of $10^3$ colony forming units (CFUs) (Fig. 2d) during treatment, by day 13 *C. difficile* levels exceeded pretreatment levels in the VAN group as the animals started relapsing. The high *C. difficile* bacterial load (Fig. 2e) and toxin activity (Fig. 2f) in cecal content confirmed recurrence of CDI in the VAN group. By comparison, none of the animals in the EVG7 group showed clinical signs of disease post-treatment. Interestingly, the total *C. difficile* load (vegetative cells and spores) in feces during EVG7 treatment was below the limit of detection and only became detectable again at day 11 (after treatment was stopped) (Fig. 2d). Furthermore, up to the end of the study, *C. difficile* levels stayed below pretreatment levels, and were significantly lower than in the relapsing VAN group (P ≤ 0.0001). On the day of necropsy (day 14/15 for VAN,

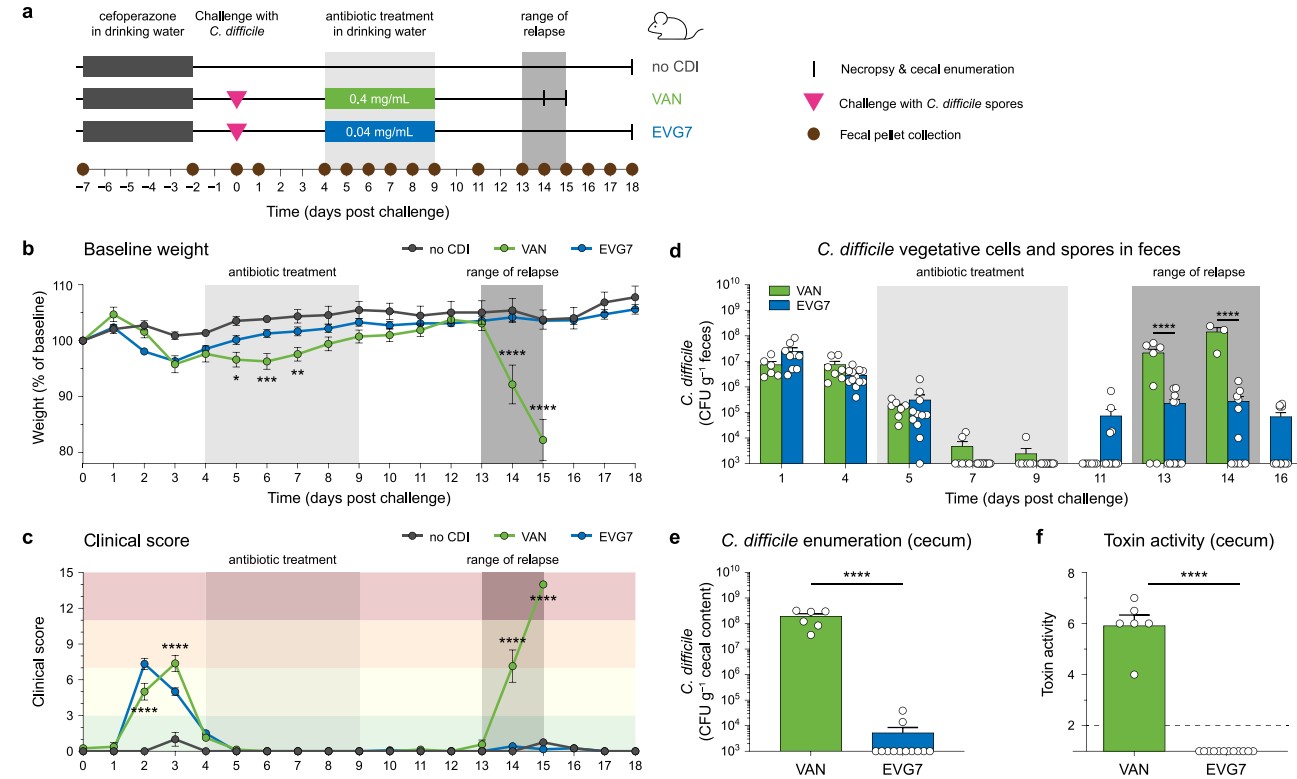

**Fig. 2 | Oral EVG7 prevents recurrence in a mouse model of rCDI. a** Timeline for in vivo model of recurrent *C. difficile* infection (rCDI) in male C57BL/6 J mice. Following cefoperazone pretreatment, the vancomycin (VAN, *n* = 8 mice) and EVG7 (*n* = 12 mice) groups were challenged with *C. difficile* via oral gavage; the 'no CDI' group (*n* = 4 mice) was not challenged with *C. difficile*. Treatment with 0.4 mg/mL vancomycin or 0.04 mg/mL EVG7 was administered in the drinking water ad libitum from days 4–9. Mice were monitored for rCDI, which occurred on days 14 and 15 in the VAN group. **b** Baseline weight loss of mice in 'no CDI' (*n* = 4), VAN (*n* = 8), and EVG7 (*n* = 12) groups. Data are mean ± standard deviation. Significance comparing VAN to EVG7 was determined by two-sided mixed-effects model analysis followed by a Tukey's posttest. **c** Clinical scoring of mice to monitor severity of CDI in the 'no CDI' (*n* = 4), VAN (*n* = 8), and EVG7 (*n* = 12) groups. Clinical score chart provided in Supplementary Table 2. Visual presentation and statistics as described for (**b**). **d** Total spores and vegetative cells of *C. difficile* in colony-forming units (CFUs) per gram of fecal content in the VAN (*n* = 8) and EVG7 (*n* = 12) groups. Each dot represents an individual fecal sample, bars represent the mean with error bars showing the standard deviation. Stool was not collected from deceased or very sick animals. Values below LOD were set to 1000. Significance and statistics as described for (**b**). **e** Total spores and vegetative cells of *C. difficile* in CFUs per gram of cecal content on day 14/15 (VAN, *n* = 6) or day 18 (EVG7, *n* = 12) (*P* < 0.0001). Each dot represents an individual cecal sample, bars represent the mean with error bars showing the standard deviation. Values below LOD were set to 1000. Significance was determined by a two-sided Mann-Whitney test. **f** Toxin activity in cecal content as determined by Vero cell cytotoxicity assay (*P* < 0.0001). Visual presentation and statistics as described for (**e**). The dashed line represents the limit of detection for the toxin activity (reciprocal log titer). For all statistical tests: *$P \le 0.05$, **$P \le 0.01$, ***$P \le 0.001$, ****$P \le 0.0001$. Graphs were generated in GraphPad Prism 10 and further compiled in Adobe Illustrator 2024. Source data are provided as a Source Data file.

day 18 for EVG7), the cecal contents of EVG7-treated mice contained a significantly lower *C. difficile* load (*P* ≤ 0.0001) (Fig. 2e) and toxin activity (*P* ≤ 0.0001) (Fig. 2f) compared to the vancomycin-treated mice. Altogether, this study indicates that 0.04 mg/mL EVG7 effectively treated the primary CDI while preventing recurrence.

To assess the impact of EVG7 dose level, we also ran the rCDI mouse model using high (0.4 mg/mL) and low (0.04 mg/mL) doses of EVG7 and vancomycin (Supplementary Fig. 1). Mice were divided into five groups of four animals (Supplementary Fig. 1a): a control group ('no CDI') that was not challenged with *C. difficile*; and four groups that received antibiotic in their drinking water from day 4–9: 0.04 mg/mL vancomycin (VAN low); 0.4 mg/mL vancomycin (VAN high); 0.04 mg/mL EVG7 (EVG7 low); and 0.4 mg/mL EVG7 (EVG7 high). All animals were euthanized on day 14, when the first group (VAN low) relapsed (Supplementary Fig. 1b), to allow for direct comparison of *C. difficile* burden and toxin activity in cecal content across all groups. During antibiotic treatment, both groups treated with 0.04 mg/mL antibiotic saw a reduction in fecal *C. difficile* levels. *C. difficile* burden remained low for 0.04 mg/mL EVG7-treated mice (EVG7 low) after discontinuation of treatment, whereas the mice treated with the same dose of vancomycin (VAN low) showed clear clinical signs of disease and recurrence upon

treatment cessation (Supplementary Fig. 1b, c). This was further confirmed by high cecal *C. difficile* burden (Supplementary Fig. 1d) and toxin activity (Supplementary Fig. 1e) detected for the VAN low group. Compared to low dose VAN and EVG7 groups, treatment with 0.4 mg/mL vancomycin (VAN high) or EVG7 (EVG7 high) resulted in further lowering of fecal *C. difficile* levels (below detection limit) while drinking water was supplemented with either antibiotic. However, by day 14 *C. difficile* was detectable in feces for both high dose groups (Supplementary Fig. 1c). Interestingly, evaluation of cecal *C. difficile* burden (Supplementary Fig. 1d) and cecal toxin activity (Supplementary Fig. 1e) indicates that high dose EVG7 is less effective than low dose EVG7, while the opposite is true for the vancomycin groups. The finding that low dose EVG7 is superior to high dose EVG7 speaks to the possibility that above a certain threshold concentration EVG7 may begin to also eliminate commensal bacteria associated with protection against *C. difficile* colonization and relapse (vide infra).

## Low dose oral EVG7 targets *C. difficile* while sparing gut microbes from the *Lachnospiraceae* family

To define changes in the gut microbiome composition between 0.4 mg/mL vancomycin (VAN) and 0.04 mg/mL EVG7 treatment

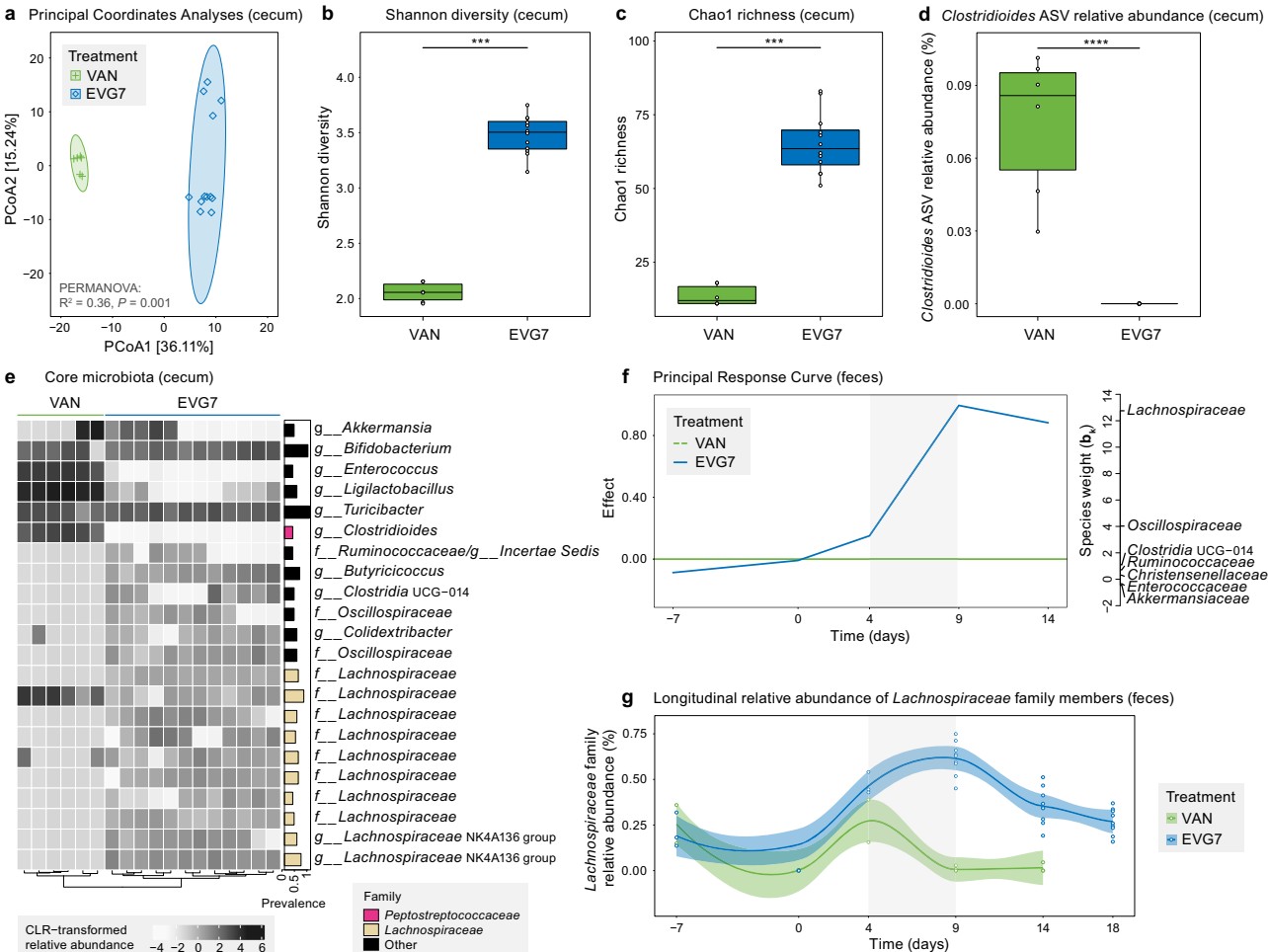

**Fig. 3 | Oral EVG7 spares the gut microbiota in a mouse model of rCDI.** Microbiome analysis of DNA extracted from cecal content samples (**a**–**e**)) of 0.4 mg/mL vancomycin-treated (VAN) mice (on day 14/15, n = 6 mice) and 0.04 mg/mL EVG7-treated mice (on day 18, n = 12 mice) or from fecal samples collected on various days (**f**, **g**)). Data accompanying Fig. 2. **a** Principal Coordinates Analysis (PCoA) plot based on Aitchison's distances between cecal content samples. The variance contribution of the first and second principal coordinates is shown on the x- and y-axis, respectively. Each point represents the microbial community of an individual sample. Ellipses correspond to 95% confidence regions. The significance of the clustering (P = 0.001) was calculated with permutational multivariate analysis of variance (PERMANOVA). **b** Shannon diversity in cecal content (P = 0.0001). Significance was determined by a two-sided Mann-Whitney test. Box plots show the median (center line), the first and third quartiles (bounds of the box). Whiskers to show the 1.5 of the IQR. Asterisks indicate statistical significance: ***P ≤ 0.001; ****P ≤ 0.0001. Each dot represents an individual cecum sample. **c** Chao1 richness in

cecal content (P = 0.0009). Visual presentation and statistics as described for (**b**). **d** Relative abundance of the *C. difficile* ASV in cecal content (P = 0.0001). Visual presentation and statistics as described for (**b**). **e** Heatmap representing the core ASVs detected in the cecal content. The core is defined by a 0.01 detection and a 0.3 prevalence threshold. The heatmap color represents centered log-ratio (CLR) transformed ASV relative abundances. The bar plot insert depicts the prevalence of the ASV. Treatments are clustered based on hierarchical clustering using Euclidean distance. **f** Principal response curve showing time-dependent antibiotic treatment effects in feces of VAN (reference line) and EVG7 groups. Absolute abundances were log10 transformed. Taxa with $b_k$ values above 0.3 or below −0.3 are displayed. **g** Longitudinal relative abundance of *Lachnospiraceae* family members in feces. Each point represents an individual stool sample. The measure of center of the regression line is fitted with a smooth local regression (LOESS), error bands indicate a 95% confidence interval. Graphs were generated in R[99] v4.1.2 and further compiled in Adobe Illustrator 2024. Source data are provided as a Source Data file.

groups (Fig. 2), DNA isolated from the mouse cecal content was characterized by 16S rRNA Illumina sequencing (Fig. 3). Principal Coordinates Analysis (PCoA) revealed that the gut microbiota profiles of each antibiotic treatment formed a distinct cluster with significant dissimilarity (P = 0.001) to the other treatment group (Fig. 3a). One of the hallmarks of a healthy gut microbiome is a high microbial diversity and richness[37]. Compared to the relapsing vancomycin-treated mice, the cecal content in the EVG7-treated mice had a significantly higher Shannon diversity (Fig. 3b; P = 0.0001) and Chao1 richness (Fig. 3c; P = 0.0009). Furthermore, the lower relative abundance of the *Clostridioides* amplicon sequence variant (ASV) in EVG7-treated mice compared to vancomycin-treated mice (Fig. 3d; P = 0.0001) is in agreement with the lower *C. difficile* CFUs and toxin activity in cecal content (Fig. 2e, f). Specifically, an increased abundance of ASVs

belonging to *Lachnospiraceae* and *Bifidobacteriaceae* family members was found in the core microbial cecal content of EVG7-treated mice compared to the vancomycin group, whereas ASVs of *Lactobacillaceae* and *Enterococcaceae* were overrepresented in the vancomycin-treated mice (Fig. 3e). Members of the *Lachnospiraceae* family are associated with resistance against *C. difficile* colonization and are among the gut microbiota that are considered protective against CDI[38–40] whereas *Lactobacillaceae* and *Enterococcaceae* are among the taxa positively correlated with *C. difficile* colonization[38]. Mouse fecal samples also reveal obvious changes in microbial community structures during and after treatment with vancomycin or EVG7 (Fig. 3f, g). Principal response curve analysis (Fig. 3f) showed time-dependent antibiotic treatment effects and revealed that the EVG7-treated mice start to deviate from the vancomycin-treated mice during treatment.

Specifically, the negative taxon weights of *Enterococcaceae* and *Akkermansiaceae* suggest a decreased absolute abundance in the EVG7-treated mice compared to the vancomycin-treated mice. Conversely, the absolute abundance of *Oscillospiraceae*, *Clostridia UCG-014*, and the protective taxa *Lachnospiraceae* and *Ruminococcaceae* increased in response to EVG7 treatment, with *Lachnospiraceae* mainly driving microbiota changes due to antibiotic treatment. The longitudinal analysis (Fig. 3g) revealed a similar increase in relative abundance of *Lachnospiraceae* in feces during and after EVG7 treatment, whereas a strong decrease was observed during vancomycin treatment. Taken together, these data in EVG7-treated mice indicate preservation of *Lachnospiraceae* family members who are associated with protection against *C. difficile* colonization in the gut.

### EVG7 exhibits an activity spectrum supportive of *C. difficile* eradication while preserving commensal gut bacteria

To further characterize the different impact of EVG7 versus vancomycin on the gut microbiome, their in vitro antimicrobial activities were evaluated against a panel of clinical commensal anaerobes comprised of representative Gram-positive and Gram-negative species (Fig. 4). Notably, the *Lachnospiraceae* family members that were identified in the mouse microbiome (Fig. 3) do not have cultured isolates in public repositories and we therefore included a number of unclassified *Lachnospiraceae* in the panel. Among the Gram-positive clinical isolates screened, the MIC values for EVG7 were typically four-to eight-fold lower than for vancomycin (Fig. 4, Supplementary Table 3). All *Clostridiaceae* species (*C. hylemonae*, *C. ramosum*, and *C. scindens*) demonstrated a four- to eight-fold higher sensitivity to EVG7, mirroring the susceptibility of *Clostridioides*. Inhibition of commensal Clostridial bacteria may negatively affect clinical outcome as specific strains have been associated with (bile acid-dependent and -independent) resistance to CDI[41–45]. However, in our studies *C. difficile* (MIC = 0.125–0.25 μg/mL) is more sensitive to EVG7 than *C. scindens* (MIC = 0.5 μg/mL) and *C. ramosum* (MIC = 1 μg/mL). Whether inhibition of commensal could affect clinical outcomes warrants further investigation. In contrast, the EVG7 MICs measured for the *Lachnospiraceae* isolates were only two- to four-fold lower than the vancomycin MICs. Interestingly, in case of vancomycin, the *Lachnospiraceae* (MIC = 0.25–0.5 μg/mL) were found to be consistently more sensitive than the *C. difficile* isolates (MIC = 1 μg/mL), an effect not observed for EVG7. These findings indicate that upon treatment with vancomycin, *Lachnospiraceae* species will likely be lost before *C. difficile* is impacted.

Among the Gram-negative commensal isolates tested, all were resistant to vancomycin (MIC ≥ 32 μg/mL) (Fig. 4, Supplementary Table 3). Gram-negative bacteria are intrinsically resistant to vancomycin, because it is unable to penetrate the outer membrane, preventing access to its target lipid II, located on the inner membrane in the periplasm[46]. Approaches to potentiate vancomycin (derivatives) against Gram-negative bacteria include disruption of the outer membrane[47,48] or conjugation of cationic residues[49–51] to facilitate translocation across the outer membrane. In the case of EVG7, most Gram-negative commensals were also resistant to EVG7 (MIC > 16 μg/mL) with some notable exceptions. Specifically, three commensal anaerobic Gram-negative isolates were found to be susceptible to EVG7; ranging from the moderately sensitive *A. onderdonkii* (MIC = 8 μg/mL) to the more highly sensitive *P. clara* (MIC = 0.25 μg/mL) and *A. muciniphila* (MIC = 0.25 μg/mL).

### Discussion

Oral vancomycin effectively treats active *C. difficile* infection (CDI) yet is often associated with recurrence of CDI that is attributed to continued disruption of the indigenous gut microbiota[13]. Recurrence is generally lower in fidaxomicin-treated patients, an effect that may stem from inhibition of spore formation[52] and sparing protective

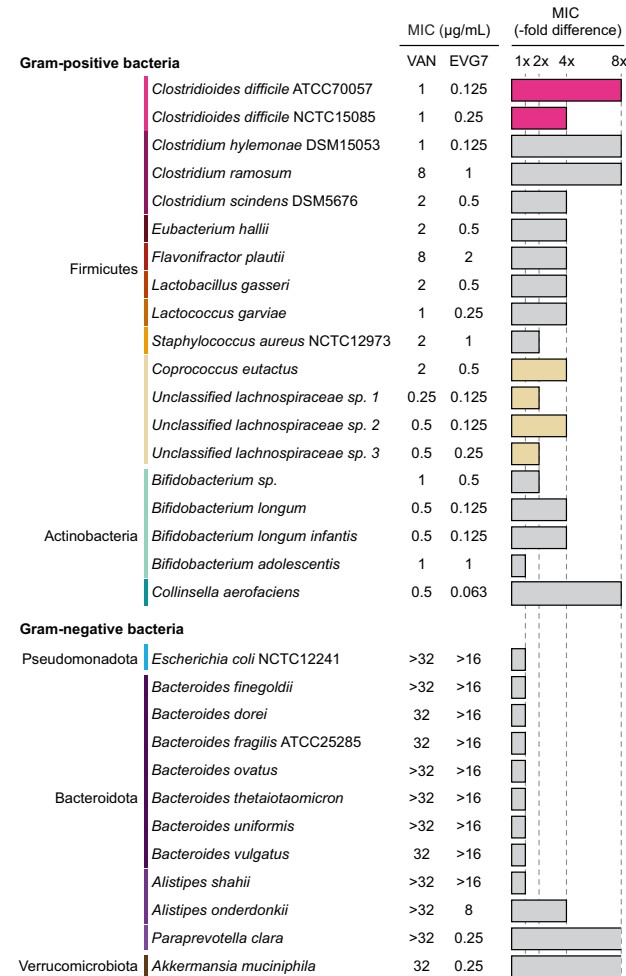

| | MIC (μg/mL) | | MIC (-fold difference) | | | |
|---|---|---|---|---|---|---|
| **Gram-positive bacteria** | VAN | EVG7 | 1x | 2x | 4x | 8x |
| *Clostridioides difficile* ATCC70057 | 1 | 0.125 | | | | |
| *Clostridioides difficile* NCTC15085 | 1 | 0.25 | | | | |
| *Clostridium hylemonae* DSM15053 | 1 | 0.125 | | | | |
| *Clostridium ramosum* | 8 | 1 | | | | |
| *Clostridium scindens* DSM5676 | 2 | 0.5 | | | | |
| *Eubacterium hallii* | 2 | 0.5 | | | | |
| *Flavonifractor plautii* | 8 | 2 | | | | |
| *Lactobacillus gasseri* | 2 | 0.5 | | | | |
| *Lactococcus garviae* | 1 | 0.25 | | | | |
| *Staphylococcus aureus* NCTC12973 | 2 | 1 | | | | |
| *Coprococcus eutactus* | 2 | 0.5 | | | | |
| *Unclassified lachnospiraceae sp. 1* | 0.25 | 0.125 | | | | |
| *Unclassified lachnospiraceae sp. 2* | 0.5 | 0.125 | | | | |
| *Unclassified lachnospiraceae sp. 3* | 0.5 | 0.25 | | | | |
| *Bifidobacterium sp.* | 1 | 0.5 | | | | |
| *Bifidobacterium longum* | 0.5 | 0.125 | | | | |
| *Bifidobacterium longum infantis* | 0.5 | 0.125 | | | | |
| *Bifidobacterium adolescentis* | 1 | 1 | | | | |
| *Collinsella aerofaciens* | 0.5 | 0.063 | | | | |
| **Gram-negative bacteria** | | | | | | |
| *Escherichia coli* NCTC12241 | >32 | >16 | | | | |
| *Bacteroides finegoldii* | >32 | >16 | | | | |
| *Bacteroides dorei* | 32 | >16 | | | | |
| *Bacteroides fragilis* ATCC25285 | 32 | >16 | | | | |
| *Bacteroides ovatus* | >32 | >16 | | | | |
| *Bacteroides thetaiotaomicron* | >32 | >16 | | | | |
| *Bacteroides uniformis* | >32 | >16 | | | | |
| *Bacteroides vulgatus* | 32 | >16 | | | | |
| *Alistipes shahii* | >32 | >16 | | | | |
| *Alistipes onderdonkii* | >32 | 8 | | | | |
| *Paraprevotella clara* | >32 | 0.25 | | | | |
| *Akkermansia muciniphila* | 32 | 0.25 | | | | |

Firmicutes; Actinobacteria; Pseudomonadota; Bacteroidota; Verrucomicrobiota (taxonomic groupings indicated at left)

**Fig. 4 | Relative to *C. difficile*, commensal bacteria are more sensitive to vancomycin than to EVG7.** Minimal Inhibitory Concentration (MIC) of vancomycin (VAN) and EVG7 against commensal clinical isolates determined by agar dilution method on Fastidious Anaerobe Agar with Horse Blood (FAA-HB)[16]. All isolates in this panel have been obtained from public repositories (as indicated by their catalog number) or isolated from human donors at the Leiden University Medical Center (LUMC) in the Netherlands. Source data are provided in Supplementary Table 3.

commensal members of the anaerobic gut microbiota[53,54]. However, similar recurrence rates have been reported in patients treated with fidaxomicin or vancomycin for infections caused by the hypervirulent BI/NAP1/027 strain[55]. Unlike fidaxomicin, sub-MIC concentrations of vancomycin do not significantly affect *C. difficile* sporulation[52]. Similarly, EVG7 (at 0.125× MIC) does not significantly reduce spore formation in *C. difficile* strain 630Δ*erm*[56,57] (data not shown). These findings indicate that EVG7 does not prevent recurrence of CDI by inhibition of sporulation, but by sparing of members of the protective gut microbiota (vide infra). As a last resort treatment after multiple recurrences, fecal microbiota transplantation (FMT) – the introduction of fecal matter from healthy donors into the patients' gastrointestinal tract to restore a healthy gut microbiome – is used for CDI management with reported recurrence prevention of 80–90%[58,59]. Given that FMT products are manufactured from human stool, they have a variable donor-dependent composition[60] and carry a risk of inadvertent transplantation of pathogenic bacteria[61,62]. Despite these risks, FMT is the only non-antibiotic treatment option for pediatric rCDI patients[31] and patients with fulminant CDI[32,63]. However, with the leading US stool bank OpenBiome suspending distribution of FMT products late 2024[27,28], availability of screened donor stool for FMT has become

limited. The advent of alternative microbiota-focused therapies might overcome these difficulties[21,22,24]. Nevertheless, these treatments do not replace antibiotic therapy as they require a neoadjuvant antibiotic course[23]. Also, while the FDA-approved VOWST[29] and REBYOTA[30] are indicated for adult rCDI, neither have been approved for treatment of fulminant CDI or pediatric patients, leaving an urgent gap in treatment.

There is a clear need for antibiotics that support microbiome conservation and stewardship – i.e. that spare the indigenous gut microbiome[64]. Disruption of the gut microbiome renders the patient susceptible to subsequent microbial infections[65] and may also affect host physiology and immune function[66] with altered gut microbiome composition found in various diseases[37,67]. Despite the clear clinical need, the pipeline for novel antibiotics that treat and prevent rCDI while sparing the gut microbiome is relatively dry[18,68,69]. The WHO reported only five traditional antibiotics in clinical development for CDI in their 2023 report[70] and development of ridinilazole (SMT19969)[71,72] and oxaquin (DNV3837)[73] has since been discontinued. The remaining three drug candidates in active development all prevented CDI recurrence in Phase 2 trials, but only CRS3123 (REP 3123)[74,75] and ibezapolstat (ACX-362E)[76] showed preservation of intestinal microbiota compared to vancomycin, while recurrence prevention by MGB-BP-3[77] was attributed to its fast bactericidal effect rather than sparing the gut microbiome. Based on reports describing the in vitro activity of these clinical candidates[78,79], EVG7 appears to possess superior activity against *C. difficile*. The reported in vitro potency against *C. difficile* clinical isolates for CRS3123 (MIC 0.5–1 μg/mL)[78] and ibezapolstat (MIC 1–8 μg/mL)[79] is similar to vancomycin (MIC 0.5–2 μg/mL), while EVG7 (MIC 0.063–0.25 μg/mL) is more potent (Fig. 1). Similar to CRS3123[74,80] and ibezapolstat[81,82], EVG7 spares beneficial gut commensal bacteria, most notably members of the *Lachnospiraceae* and *Oscillospiraceae* families. In all cases, new antibiotic therapies for CDI will be compared to the standard of care vancomycin. In our current work, we demonstrate that oral administration of the new experimental glycopeptide EVG7 effectively treats primary CDI in a murine model (Fig. 2). When dosed orally at 0.04 mg/mL, EVG7 was more effective than a 10-fold higher dose of oral vancomycin in the prevention of CDI relapse, thus demonstrating superiority over the standard of care.

Protection against recurrent CDI requires sufficient preservation of the gut microbiome during antibiotic treatment. The *Lachnospiraceae* are a family of obligate anaerobic Gram-positive bacteria belonging to the Clostridia cluster XIVa[83] and are among the microbial taxa associated with protection against *C. difficile* in both mice and humans[38–40]. Our data (Fig. 3) indicate that treatment with a low dose of oral EVG7 (0.04 mg/mL) prevents disease recurrence in a murine model of rCDI by the sparing of *Lachnospiraceae*. Mice colonized by *Lachnospiraceae* are less likely to develop CDI[38] and administration of specific murine *Lachnospiraceae* isolates has also been shown to protect against *C. difficile* colonization in murine CDI models[84,85]. In humans, the presence of *Lachnospiraceae* was negatively correlated with *C. difficile* colonization in both healthy adults and fecal transplant patients[38]. Furthermore, the protective potential of *Lachnospiraceae* for treatment of rCDI is exploited in multiple microbiota-focused products currently approved or in clinical trials[86] with 36% of the genera in the recently FDA-approved VOWST being *Lachnospiraceae*[87]. In this regard, the maintenance of *Lachnospiraceae* species in the context of CDI treatment with low dose oral EVG7 is promising. Based on the findings here reported, EVG7 warrants further investigation as an alternative to oral vancomycin therapy in the treatment of CDI and the prevention of recurrence.

## Methods
### Materials and resources
A list of the reagents used in this study is provided in Supplementary Table 4. Bacterial isolates used in this study are available upon

reasonable request to W.K.S. Samples of the experimental antibiotic EVG7 are available upon request to N.I.M.

### Minimum inhibitory concentration (MIC) assays (agar dilution)
**Antibiotic preparation.** Solutions of vancomycin (Applichem, A1839) and EVG7 (Leiden University, in-house synthesis[33]) were prepared from dry solid powder. Stock solutions of vancomycin (6.4 mg/mL) and EVG7 (3.2 mg/mL) were prepared in distilled water, filter-sterilized, and stored at −20 °C until use. The final concentrations of the antimicrobials in the agar dilution experiments were 0.25–64 mg/L for vancomycin and 0.125–32 mg/L for EVG7.

**C. difficile isolates.** The *Clostridioides difficile* isolates tested belong to a well-established reference collection[88,89] supplemented with clinical isolates collected by the National Expertise Center for *C. difficile* infections (hosted at the Leiden University Medical Center (LUMC), Leiden, NL).

**Commensal isolates.** Gut microbiome commensal strains were isolated from healthy human donor feces at the Experimental Bacteriology laboratory at the LUMC (Leiden, NL). Though strains were isolated on different culture media, all demonstrated robust growth on Fastidious Anaerobe Agar supplemented with horse blood. All strains were identified by standard procedures (including Bruker Biotyper and 16S rRNA sequencing) and stored in glycerol broth at −70 °C until use.

**Agar dilution MIC assay.** Minimum inhibitory concentrations (MICs) for *C. difficile* were determined at the LUMC (Leiden, NL) using a modified Clinical and Laboratory Standard Institute (CLSI) agar dilution method as previously reported[16]. Briefly, *C. difficile* isolates were removed from −80 °C storage and subcultured anaerobically on Tryptic Soy Sheep Blood Agar plates (TSS; bioMérieux, 43009) for 48 h, prior to inoculation of pre-reduced Schaedler's anaerobic broth (ThermoFisher Oxoid, CM0497B) for 24 h. Isolates were transferred to pre-reduced sterile saline (Supelco, 1.06404.1000) and adjusted to McFarland standard 1.0. Non-antibiotic-containing plates were incubated aerobically and anaerobically. Test medium used for *C. difficile* strains (Fig. 1) was Brucella Blood Agar (BBA; ThermoFisher Oxoid, CM0169) supplemented with 5% (v/v) sheep blood (Xebios Diagnostics, 10000100/10000250), 0.1 g/L hemin (SigmaAldrich, 51280), and 0.01 g/L vitamin $K_1$ (Carl Roth, 3804.2). Minimum inhibitory concentrations (MICs) for the panel of commensal isolates (Fig. 4) were determined at the LUMC (Leiden, NL) using methods recommended by EUCAST (www.eucast.org, v15.0), using Fastidious Anaerobe Agar (FAA; Neogen, NCM0014a) supplemented with 5% (v/v) horse blood (Xebios Diagnostics, 2000100). *C. difficile* was included in these experiments as a control strain as well. Bacterial suspensions were inoculated onto agar plates using a multipoint inoculator and incubated anaerobically at 37 °C for 48 h. The minimum inhibitor concentration was defined as the lowest dilution at which growth is completely inhibited or where only a single colony remains[16].

### Murine model of recurrent *C. difficile* infection (rCDI)
**Antibiotic preparation.** Color- and odorless solutions of vancomycin (Sigma Aldrich, V2002) and EVG7 (Leiden university, in-house synthesis[33]) in drinking water (Gibco Laboratories, 15230) were prepared from solid dry powder and stored at 4 °C for a maximum of 7 days.

**Spore preparation.** *C. difficile* spores were prepared as previously described in ref. 90. Briefly, *C. difficile* was grown at 37 °C anaerobically for 1 week in Clospore media (in-house preparation[91]). Spores were harvested by centrifugation and washed with water, heat treated for 20 min at 65 °C, and were stored at 4 °C. Spores were plated on Brain Heart Infusion (BHI; BD Life Sciences, 241810) and Taurocholate BHI

(TBHI; in-house preparation[90]) to make sure no viable cells were present.

**Animals and housing.** Animal experiments were conducted in the Laboratory Animal Facilities located on the North Carolina State University (NCSU) College of Veterinary Medicine (CVM) campus (Raleigh, USA). C57BL/6 J mice (male, 4–5 weeks old) purchased from Jackson Laboratories were used for the experimental infections. Mice were housed in cages of four animals with autoclaved bedding and water, and irradiated food. Cage changes were performed weekly in a laminar flow hood. All mice were subjected to a 12-hour light and 12-hour dark cycle, with an average temperature of 70 °F and 35% humidity.

**C. difficile infection and sample collection.** Groups of 4–5-week-old, male C57BL/6 J mice (Jackson Laboratories, 664) were given 0.5 mg/mL cefoperazone (MP Biomedicals, 02199695-CF) in drinking water ad libitum for 5 days to render them susceptible to *C. difficile* infection[90]. This was followed by a 2-day washout with regular drinking water (Gibco Laboratories, 15230). On day 0 all mice, excluding the'no CDI' control group, were challenged via oral gavage with approximately $10^5$ *C. difficile* 630 spores. All mouse stool tested culture negative for *C. difficile* before the challenge. From days 4–9, the mice were treated as follows based on their experimental group in study I (Fig. 2) and study II (Supplementary Fig. 1):

   (1) 'no CDI' uninfected control ($n = 4$ in study I; $n = 4$ in study II) – not challenged with *C. difficile* and not given any additional antibiotic on day 4;

   (2) VAN high ($n = 8$ in study I; $n = 4$ in study II) – challenged with *C. difficile*, given 0.4 mg/mL vancomycin in drinking water ad libitum;

   (3) VAN low ($n = 4$ in study II) – challenged with *C. difficile*, given 0.04 mg/mL vancomycin in drinking water ad libitum;

   (4) EVG7 high ($n = 4$ in study II) – challenged with *C. difficile*, given 0.4 mg/mL EVG7 in drinking water ad libitum;

   (5) EVG7 low ($n = 12$ in study I; $n = 4$ in study II) – challenged with *C. difficile*, given 0.04 mg/mL EVG7 in drinking water ad libitum.

   *C. difficile* infection and recurrence progression/status of the mice was monitored closely starting on day 0 by weighing the mice daily and observing and recording the clinical signs of disease for up to 18 days post challenge. An adjusted CDI clinical scoring chart (Supplementary Table 2) was used in the assessment of the clinical signs of disease in the mice and the information as used in determining severity of the disease[92]. Fecal pellets were collected on day −7, −2, 0, 1, 4, 5, 6, 7, 9, 11, 13, 14 in study I and II, and additionally on day 8, 15, 16, 17, and 18 in study I. Fecal pellets were stored at −80 °C until further analysis. Stool collection was not possible for all animals on some days, because sick animals often do not defecate: details and exceptions are provided in the 'Source Data document'. Animals were humanely euthanized by $CO_2$ asphyxiation followed by cervical dislocation prior to necropsy. Animals that expired before the end of the study were not necropsied: details are provided in the 'Source Data document'. During necropsy, content and tissue snips were taken from the cecum and flash frozen in liquid nitrogen then stored at −80 °C until further analysis. Some of the cecal content that was collected was used for bacterial enumeration on the day of necropsy, and later 16S rRNA sequencing analysis.

**C. difficile infection clinical signs scoring.** The clinical scoring chart was developed for the detection and assessment of clinical signs of disease in mice that are known to have been infected by *C. difficile*. There are five areas of typical clinical signs of CDI that are assessed for this chart: weight loss, energy level/activity, posture, coat cleanliness, and presence of diarrhea. Each area is scored from 0 to 3 according to the chart (Supplementary Table 2) adjusted from Warren et al.[92], with 0 showing no clinical signs and 3 showing severe clinical signs, all scores are added together for a full assessment. An overall score of ≤3 is not specifically indicative of signs of disease of CDI and any physical signs of sickness could be due to either stress or another condition (such as dehydration or malocclusion). An overall score of 4–7 is indicative that the mouse is showing signs of CDI but is not necessarily severely impacted by the disease. A score of 8–11 is indicative that the mouse has severe signs of disease associated with CDI and should be monitored closely and considered for euthanasia if signs appear to worsen. An overall score of ≥12 is indicative of very severe signs of disease of CDI, the mouse can often be considered moribund at this level, and euthanasia should be strongly considered. If the mouse reaches at least 20% weight loss from the day 0 baseline weight at any time then it has reached the humane endpoint in accordance with the protocol and should be euthanized.

**Bacterial enumeration of C. difficile in stool and cecal content**
Bacterial enumeration was performed at NCSU (Raleigh, USA) on both fecal pellets and cecal content as reported in Winston et al.[90]. Feces were diluted (1:10 based on weight) with phosphate-buffered saline (PBS; ThermoFisher, 10-010-049) to resuspend the samples. Samples were then incubated anaerobically for 30 min to allow the contents to settle. Ten-fold serial dilutions were performed for each sample in 1× PBS and was plated on the *C. difficile*-selective medium: Taurocholate, Cefoxitin, Cycloserine, and Fructose Agar (TCCFA; preparation as previously reported[90]) to determine the total number of both vegetative cells and spores present. All agar plates were incubated at 37 °C for 24 h. Bacterial load was expressed as colony-forming units CFUs/gram of feces or cecal content. Samples with burden below limit of detection (LOD) were given a value of $10^3$ CFU/g for graphical and statistical purposes.

**Vero cell cytotoxicity assay**
Toxin activity was measured at NCSU (Raleigh, USA) using a Vero cell cytotoxicity assay as reported in Winston et al.[90]. Vero cells are grown and maintained in DMEM media (Gibco laboratories, 11965-092) with 10% heat-inactivated fetal bovine serum (FBS; Gibco Laboratories, 16140-071) and 1% penicillin-streptomycin solution (Gibco Laboratories, 15070-063). Cells were incubated with 0.25% trypsin (Gibco Laboratories, 25200-056), washed with DMEM media, and harvested by centrifugation at 250 × g for 5 min. Plates were seeded at $1 × 10^4$ cells per well in a 96-well flat-bottom microtiter plate (Costar Corning, CL3595) and incubated overnight at 37 °C/5% $CO_2$. The cecal content was diluted (1:10 by weight) with PBS and filtered before testing. The samples were further diluted 10-fold to a maximum of $10^{-6}$. Sample dilutions were incubated 1:1 with PBS (for all dilutions) or antitoxin (TechLabs, T5000) for 40 min at room temperature. Following the incubation, these mixtures were added to the Vero cells and plates were incubated overnight at 37 °C/5% $CO_2$. Vero cells were viewed under 200× magnification and evaluated for rounding after overnight incubation. The cytotoxic titer was defined as the reciprocal log of the highest dilution that produced rounding in 80% of Vero cells for each sample. Values below LOD were set to 1 for graphical and statistical purposes. Vero cells treated with purified *C. difficile* toxin A (List Biological Labs, 152 C) were used as controls.

**16S rRNA Illumina sequencing**
DNA was isolated from feces and cecal snips (from study I) at the University of Michigan Microbial Systems Molecular Biology Laboratory (Ann Arbor, USA). The Mag Attract Power Microbiome kit (Mo Bio Laboratories, Inc) was used to isolate DNA from cecal snips. Dual-indexing sequencing approach was used to amplify the V4 region of the 16S rRNA gene. Each PCR mixture contained 2 μL of 10× Accuprime PCR buffer II (Life Technologies, CA, USA), 0.15 μL of Accuprime high-fidelity *polymerase* (Life Technologies, CA, USA), 5 μL of a 4.0 μM primer set, 1 μL DNA, and 11.85 μL sterile nuclease-free water. The template DNA concentration was 1–10 ng/μL for a high bacterial DNA/host DNA ratio. The PCR conditions were as follows: 2 min at 95 °C, followed

by 30 cycles of 95 °C for 20 s, 55 °C for 15 s, and 72 °C for 5 min, followed by 72 °C for 10 min. Libraries were normalized using a Life Technologies SequalPrep normalization plate kit (Life Technologies, CA, USA) as per the manufacturer's instructions for sequential elution. The concentration of the pooled samples was determined using the Kapa Biosystems library quantification kit for Illumina platforms (Kapa Biosystems, MA, USA). Agilent Bioanalyzer high-sensitivity DNA analysis kit (Agilent, CA, USA) was used to determine the sizes of the amplicons in the library. The final library consisted of equal molar amounts from each of the plates, normalized to the pooled plate at the lowest concentration. Sequencing was done on the Illumina MiSeq platform, using a MiSeq reagent kit V2 (Ilumina, CA, USA) with 500 cycles according to the manufacturer's instructions, with modifications. Sequencing libraries were prepared according to Illumina's protocol for preparing libraries for sequencing on the MiSeq (Ilumina, CA, USA) for 2 or 4 nM libraries. PhiX and genomes were added in 16S amplicon sequencing to add diversity. Sequencing reagents were prepared according to the Schloss SOP, and custom read 1, read 2 and index primers were added to the reagent cartridge. FASTQ files were generated for paired end reads.

### Microbiome data analysis

**Taxonomic profiling.** The demultiplexed 16S rRNA gene amplicon sequences were filtered using "trimmomatic"[93] v0.36. Filtered sequences were processed with QIIME 2[94] v2024.10 to produce amplicon sequence variants (ASVs). In brief, reads were denoised with the DADA2 plugin[95] and taxonomies were assigned against the SILVA 138 database using the 99% OTUs full-length sequences Naive Bayes classifier with the classify-sklearn algorithm in the feature-classifier plugin[96–98]. Further processing was done using R[99] v4.1.2.

**PCoA.** The base R "cmdscale" function and the R "phyloseq" package[100] v1.38.0 were used to perform and visualize PCoA ordination with Aitchison's distance of cecum samples. Euclidean distance was calculated for CLR-transformed ASV read counts with the "phyloseq" package[100] v1.38.0.

**Microbiota ecological characteristics.** The "alpha" function of the R package "microbiome"[101] v1.16.0 was used to calculate Shannon diversity and Chao1 richness based on rarified ASV abundance profiles of each sample. Samples were rarified to an even depth (minimum number of reads detected) with the R "phyloseq" package[100] v1.38.0.

**Heatmap.** The core was determined on ASV relative abundance profiles with the "core" function in the R "microbiome" package v1.16.0 with a 0.01 detection threshold and a 0.3 prevalence threshold. Core ASV relative abundances were CLR-transformed and visualized with the "comp_heatmap" function of the R "microViz" package[102] v0.9.3.

**Principal response curve.** Principal response curve analysis was performed on log10-transformed absolute abundances of families using the "prc" function of the R package "vegan"[103] v2.6-4 using symmetric scaling. Stool samples from day 18 of the EVG7-treated mice were excluded from this analysis. The R "ggvegan" package[104] v0.1.999 was used for visualization.

**Longitudinal analyses.** The regression line for relative abundance of *Lachnospiraceae* family members on various days was plotted for vancomycin-treated mice and EVG7-treated mice with the "geom_smooth" function of the "ggplot2" package[105] v3.3.6 using the default smooth local regression (LOESS).

### Statistical analyses

Statistical analyses of in vivo study data (Fig. 2 and Supplementary Fig. 1) were performed using GraphPad Prism 10 v.10.3.1. For burden data, CFU values (Fig. 2d, e and Supplementary Fig. 1c, d) were transformed to log10 values for statistical analysis. Significance comparing VAN to EVG7 for baseline weight, clinical score and fecal *C. difficile* CFU (Fig. 2b–d) was determined by a two-sided mixed-effects model analysis followed by a Tukey's posttest. Significance for cecal *C. difficile* CFU and cecal toxin activity (Fig. 2e, f) was determined by a two-sided Mann-Whitney test. Significance for cecal *C. difficile* CFU (Supplementary Fig. 1d) was determined by a one-sided Brown-Forsythe and Welch's ANOVA followed by a Dunnett T3 multiple posttest. Significance for toxin activity (Supplementary Fig. 1e) was determined by a one-sided Kruskal-Wallis ANOVA followed by a Dunn's multiple comparisons posttest. For all tests: *$P \leq 0.05$, **$P \leq 0.01$, ***$P \leq 0.001$, ****$P \leq 0.0001$. Details are also listed in each figure legend.

Statistical analyses of microbiome data (Fig. 3) were performed using R[99] v4.1.2. Significance of clustering by treatment in the PCoA (Fig. 3a) was tested with the "adonis2" function using default permutations in the R "vegan" package[103] v2.6-4. Means of Shannon diversity (Fig. 3b), Chao1 richness (Fig. 3c), and *Clostridioides* ASV relative abundance (Fig. 3d) were compared with a two-sided Mann-Whitney test after normality was rejected with a Shapiro-Wilk's test. The "geom_smooth" function of the "ggplot2" package[105] v3.3.6 was used for a default smooth local regression (LOESS) with 95% confidence interval in the longitudinal analyses of *Lachnospiraceae* family members (Fig. 3g).

### Ethics statement

All animal experiments were conducted in the Laboratory Animal Facilities located on the North Carolina State University (NCSU) College of Veterinary Medicine (CVM) campus (Raleigh, USA). The animal facilities are equipped with a full-time animal care staff coordinated by the Laboratory Animal Resources (LAR) division at NCSU. The NCSU CVM is accredited by the Association for the Assessment and Accreditation of Laboratory Animal Care International (AAALAC). Trained animal handlers in the facility fed and assessed the status of animals several times per day. Those assessed as moribund were humanely euthanized by $CO_2$ asphyxiation. This protocol is approved by NC State's Institutional Animal Care and Use Committee (IACUC).

Bacteria isolated from human stool were used in this study (not human stool itself). These bacteria were originally isolated from stool obtained from the Netherlands Donor Feces Bank (NDFB; https://www.ndfb.nl/), independent from the present study. The NDFB obtained informed consent for the use of the fecal donor samples for research purposes under approval of the Medical Ethics Committee at Leiden University Medical Center (P15.145).

### Reporting summary

Further information on research design is available in the Nature Portfolio Reporting Summary linked to this article.

## Data availability

The demultiplexed 16S rRNA gene amplicon sequencing data generated in this study have been deposited in the European Nucleotide Archive under accession code PRJEB86983. Source data are provided in the Supplementary Information and the Source Data file. Source data are provided with this paper.

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

## Acknowledgements
This work was funded in part by the Netherlands Organization for Scientific Research (NWO NACTAR); grants number 18504 (N.I.M.) and 20813 (N.I.M.). The authors acknowledge the Netherlands Donor Feces Bank for kindly providing a stool sample from which bacterial strains were isolated. The authors wish to acknowledge Lesley Schout for banking the isolates used for this study. Marjon Wells-Bennik and Guus Kortman (NIZO) are thanked for the early testing of EVG7 against Clostridial strains.

## Author contributions
E.M., W.K.S., C.M.T., and N.I.M. designed the study. E.M., F.M.S., and E.G. synthesized EVG7. I.M.J.G.S., A.G.R., and W.K.S. isolated clinical strains and performed in vitro susceptibility assays. C.E.P. and C.M.T. performed in vivo studies. J.E.G.H. and W.K.S. performed microbiome analysis. E.M. and J.E.G.H. prepared the first draft of the manuscript with subsequent contributions from the other authors. All authors discussed results and approved the manuscript.

## Competing interests
E.G. and N.I.M. are inventors on a patent application filing WO2021060980A1; title: "Antibiotic compounds"; priority date: 24 September 2019, which includes compounds described in this article. W.K.S. has performed research for Cubist Pharmaceuticals and Acurx Pharmaceuticals and received speaker fees from Promega. C.M.T. has consulted for Vedanta Biosciences, Inc., Summit Therapeutics, and Ferring Pharmaceuticals, Inc. and is on the Scientific Advisory Board for Ancilia Biosciences. All other authors declare that they have no competing interests.
