## [Transparent Peer Review file · Nature Communications]

Experimental glycopeptide antibiotic EVG7 prevents recurrent *Clostridioides difficile* infection by sparing members of the *Lachnospiraceae* family

Corresponding Author: Professor Nathaniel Martin

Version 0:

Reviewer comments:

Reviewer #1

(Remarks to the Author)

In this study, Mons and colleagues investigated a new glycopeptide antibiotic, EVG7, for its potential as a *Clostridioides difficile* therapeutic with fewer detrimental impacts on the microbiota. This area of research is important, as treatment options for primary *C. difficile* infection are limited, and efficacy of these treatments (vancomycin and fidaxomicin) have been decreasing over time. In this study, the authors observed that EVG7 has low MIC against a broad range of clinical *C. difficile* isolates, was more efficacious in limiting recurrent infection in a mouse model of infection than vancomycin, and had limited impacts on *Lachnospiraceae* family members in the microbiota. This work builds upon previous studies completed to test efficacy of EVG7 against *S. aureus*, which also demonstrated efficacy to vancomycin with limited toxicity when administered at supratherapeutic doses. Based on these studies, EVG7 efficacy is similar to two other antibiotics currently undergoing evaluation in human clinical trials for *C. difficile* treatment (CRS3123 and ibezapolstat) that are also reported to be microbiome-sparing.

Overall, the data presented supports the conclusions drawn from the study. However, there were a few topics that should have been addressed more thoroughly in the discussion.

1. In addition to inhibiting *C. difficile*, the authors showed that EVG7 inhibited other *Clostridiaceae* in the microbiota. Recent studies have pointed to the importance of commensal *Clostridiaceae* in inhibiting *C. difficile* through multiple mechanisms (e.g., doi: 10.1038/nature13828, doi: 10.1016/j.chom.2021.09.007, doi: 10.1016/j.chembiol.2018.10.003, doi: 10.1016/j.chom.2024.12.002). The authors should discuss potential limitations of this antibiotic considering its effects on other *Clostridiaceae* in the microbiota.

2. A more thorough discussion of the relative efficacy of EVG7 compared to other antibiotics currently in clinical trials would assist the reader in better evaluating potential clinical impacts of this antibiotic.

There were also additional minor comments that could improve data presentation.

1. Please provide justification for completion of studies only in male mice.

2. Please provide R2 or F-statistic to accompany PERMANOVA analysis in Fig. 3a.

3. Fig. 3e. It is surprising that there are no *Bacteroides* species in the core microbiota. Were the *Bacteroides* absent, or were there no ASVs that met the thresholds for defining the core microbiota.

Overall, the methodology was sound and there was sufficient detail provided to allow studies to be repeated. One point that should be discussed further was the variation observed between the two studies presented in Figure 1 and Supplementary Figure 1.

Reviewer #2

(Remarks to the Author)

This paper describes the testing of a new antibiotic compared to the standard of care antibiotic, vancomycin, for the treatment of *C. difficile* infections and prevention of disease recurrence in mice. Mons et al show that clinically relevant strains of *C. difficile* are more susceptible to a new glycopeptide antibiotic (EVG7), when compared to vancomycin in vitro. Excitingly, this result translated to in vivo studies, where mice that received a low dose (0.04mg/ml) of EVG7 prevented disease recurrence in mice, whereas mice that were treated with a typical dose (0.4mg/ml) of vancomycin succumbed to disease relapse. Furthermore, upon subsequent analysis, it was found that mice treated with EVG7 retained more species

diversity than vancomycin treated mice, with preservation of members of the Lachnospiraceae family observed in EVG7-treated mice. Additionally, when tested in vitro, the MIC values for EVG7 were lower than for vancomycin when tested against members of the Lachnospiraceae family, with the Lachnospiraceae shown to be more sensitive to vancomycin than *C. difficile*. As Lachnospiraceae has previously been associated with protection against *C. difficile*, the authors conclude that EVG7 prevents disease recurrence by the sparing of Lachnospiraceae.

In addition, EVG7 (0.04mg/ml) performed better than vancomycin (0.4mg/ml) in treatment of primary disease, as although both antibiotics prevented mice from succumbing to infection during treatment, EVG7 reduced *C. difficile* shedding in the faeces below the limit of detection, whereas vancomycin did not. Furthermore, mice treated with vancomycin lost more weight during the treatment period compared to EVG7-treated mice, further highlighting the superiority of EVG7 to vancomycin in treatment of primary *C. difficile* infections.

This is a very solid piece of work and the manuscript has been beautifully drafted. Publishing this work will contribute strongly to the *C. difficile* field. We only have one suggestion that we feel would add to or strengthen the conclusion (see below).

This work will be of significance to the field because the development of a new antibiotic that more selectively targets *C. difficile*, but spares protective members of the gut microbiota meets an urgent unmet need for the development of new therapeutics for the treatment of *C. difficile* infections, and most excitingly, the prevention of disease recurrence. This is also of importance to related fields and for clinical practice because an unintended consequence of the use of vancomycin for treatment of *C. difficile* infections is that it can select for resistance in other bacterial species and drive the increased colonisation of patients with pathogenic strains of vancomycin-resistant enterococci (VRE) or *S. aureus* (VRSA).

The study is very thorough and most of the conclusions drawn are supported by the data. There is one conclusion however that we feel could be strengthened by the addition of an in vitro study. The authors conclude that EVG7 prevents disease recurrence by the sparing of Lachnospiraceae, however it is also possible that recurrence rates can be reduced if an antibiotic targets the sporulation pathway. While the authors report a reduction in total cell numbers in the faeces of mice (which encompasses both spores and vegetative cells) during EVG7 treatment, it would be interesting to see if EVG7 is able to prevent spore formation in *C. difficile*. Can the authors perform an in vitro study to examine if sub-MIC concentrations of EVG7 (that does not impact vegetative cell numbers) result in a decrease in spore formation? If so, that would strengthen the use of EVG7 as a treatment for recurrent *C. difficile* infections as it would act via two mechanisms to do this – both reducing spores and sparing Lachnospiraceae. If it doesn't impact sporulation, then it strengthens the authors conclusions that recurrence is prevented by the sparing of Lachnospiraceae.

The data has been thoroughly analysed and correctly interpreted.

The methodology is sound and meets the expectations of the field. Nature communications state that the title and/or abstract must indicate when findings apply to only one sex or gender. As this study was performed on Male mice, can the authors please add the gender of the mice to the abstract.

The described methodology is very thorough. The work could be reproduced from the detail provided.

Reviewer #3

(Remarks to the Author)

Version 1:

Reviewer comments:

Reviewer #2

(Remarks to the Author)

The changes to the manuscript are appropriate and no further changes are requested.

Reviewer #3

(Remarks to the Author)

RESPONSE TO REVIEWERS

Reviewer 1:

1. In addition to inhibiting *C. difficile*, the authors showed that EVG7 inhibited other *Clostridiaceae* in the microbiota. Recent studies have pointed to the importance of commensal *Clostridiaceae* in inhibiting *C. difficile* through multiple mechanisms (e.g., doi: 10.1038/nature13828, doi: 10.1016/j.chom.2021.09.007, doi: 10.1016/j.chembiol.2018.10.003, doi: 10.1016/j.chom.2024.12.002). The authors should discuss potential limitations of this antibiotic considering its effects on other *Clostridiaceae* in the microbiota.

Author Response: *Specific strains of commensal Clostridiaceae have indeed been associated with resistance to CDI. We have therefore added the following section to the main text in line 187-192:*

*"Inhibition of commensal Clostridial bacteria may negatively affect clinical outcome as specific strains have been associated with (bile acid-dependent and -independent) resistance to CDI^{41,42,43,44,45}. However, in our studies *C. difficile* (MIC = 0.125–0.25 µg/mL) is more sensitive to **EVG7** than *C. scindens* (MIC = 0.5 µg/mL) and *C. ramosum* (MIC = 1 µg/mL). Whether inhibition of commensal could affect clinical outcomes warrants further investigation."*

2. A more thorough discussion of the relative efficacy of EVG7 compared to other antibiotics currently in clinical trials would assist the reader in better evaluating potential clinical impacts of this antibiotic.

Author Response: *We thank the reviewer for this suggestion and now include a comparison of **EVG7** and other clinical candidates in the discussion section (lines 243-248):*

*"Based on reports describing the in vitro activity of these clinical candidates^{78,79}, **EVG7** appears to possess superior activity against *C. difficile*. The reported in vitro potency against *C. difficile* clinical isolates for CRS3123 (MIC 0.5–1 µg/mL)⁷⁸ and ibezapolstat (MIC 1–8 µg/mL)⁷⁹ is similar to vancomycin (MIC 0.5–2 µg/mL), while **EVG7** (MIC 0.063–0.25 µg/mL) is more potent (Fig. 1). Similar to CRS3123^{74,80} and ibezapolstat^{81,82}, **EVG7** spares beneficial gut commensal bacteria, most notably members of the Lachnospiraceae and Oscillospiraceae families."*

Reviewer 1 - additional minor comments:

1. Please provide justification for completion of studies only in male mice.

Author Response: *We have used both male and female mice in our experiments over the past decade and have not seen any differences in disease outcome. To adhere to rigorous statistical analysis and to reduce the number of animals used in research per the 3Rs (Replacement, Reduction, and Refinement), we chose to only use male mice in this study (as noted in the method section). We have included the reasoning above in the 'Reporting Summary' for full transparency.*

2. Please provide R² or F-statistic to accompany PERMANOVA analysis in Fig. 3a.

Author Response: *The R² of the PERMANOVA analysis is 0.36. This value has been added to Fig. 3a.*

3a. Fig. 3e. It is surprising that there are no *Bacteroides* species in the core microbiota. Were the *Bacteroides* absent, or were there no ASVs that met the thresholds for defining the core microbiota.

Author Response: *No ASVs of the *Bacteroides* genus were present in stool or cecum DNA. This is a common observation in this mouse model of rCDI, as it involves pre-treatment with cefoperazone before vancomycin or **EVG7** treatment, which likely eradicates *Bacteroides* species.*

3b. Overall, the methodology was sound and there was sufficient detail provided to allow studies to be repeated. One point that should be discussed further was the variation observed between the two studies presented in Figure 1 and Supplementary Figure 1.

Author Response: *The EVG7 low dose that was used to treat mice in both Fig. 2 and Suppl Fig. 1 had very similar results between both experiments and figures. I think the reviewer is asking about the high dose VAN-treated experiments, which is in Fig. 2 and Suppl Fig. 1. The rCDI mouse model allows us to measure recurrent CDI, but there is a range of days for when this occurs. It can be anywhere from day 13 to 16 in this mouse model with the high VAN dose. In Fig. 2 we let rCDI occur in the high dose VAN group (control group for rCDI) and we euthanized mice when they met their human endpoint, weight loss in Fig. 2b, which was on day 14–15. We let the EVG7-treated mice go out further than this to evaluate if EVG7 prevented rCDI, which it did, so they were euthanized on a different day, day 18. The days of necropsy were different in Fig. 2 and Suppl Fig. 1, so there are some differences in the high dose VAN group because of this. In Suppl Fig. 1, the mice were treated with different doses of both VAN and EVG7, and then we did necropsy on the same day or when the first group met their clinical endpoint (weight loss) for rCDI, so we could compare mice across treatment on the same day. Since rCDI occurs in this mouse model during a range of day 13–16, we might have caught the high dose VAN-treated mice a day early from more severe rCDI signs of disease. If we had waited one more day to do necropsy then it would have looked like the mice in Fig. 2. Again, the question in Fig. 2 and Suppl Fig. 1 was a bit different and this is why they days of necropsy are different. This is detailed in the methods section and in the figure legends.*

Reviewer 2:

1. The study is very thorough and most of the conclusions drawn are supported by the data. There is one conclusion however that we feel could be strengthened by the addition of an in vitro study. The authors conclude that EVG7 prevents disease recurrence by the sparing of Lachnospiraceae, however it is also possible that recurrence rates can be reduced if an antibiotic targets the sporulation pathway. While the authors report a reduction in total cell numbers in the faeces of mice (which encompasses both spores and vegetative cells) during EVG7 treatment, it would be interesting to see if EVG7 is able to prevent spore formation in *C. difficile*. Can the authors perform an in vitro study to examine if sub-MIC concentrations of EVG7 (that does not impact vegetative cell numbers) result in a decrease in spore formation? If so, that would strengthen the use of EVG7 as a treatment for recurrent *C. difficile* infections as it would act via two mechanisms to do this – both reducing spores and sparing Lachnospiraceae. If it doesn't impact sporulation, then it strengthens the authors conclusions that recurrence is prevented by the sparing of Lachnospiraceae.

Author Response: *We have performed the suggested in vitro sporulation assay, and found that EVG7 (at 0.125× MIC) does not significantly reduce spore production. This is in line with findings for vancomycin (at 0.25× MIC) (Babakhani, 2012). In this study, we used the 0.125× MIC concentration for EVG7, because growth defects were still observed at 0.25× MIC. For the reviewer's consideration, we include the protocols used for this study and corresponding results on the final page of this document. In revising our manuscript we now include the following text addressing the effect of EVG7 on sporulation (lines 214-218):*

*“Unlike fidaxomicin, sub-MIC concentrations of vancomycin do not significantly affect *C. difficile* sporulation⁵². Similarly, EVG7 (at 0.125× MIC) does not significantly reduce spore formation in *C. difficile* strain 630Δerm^{56,57} (data not shown). These findings indicate that EVG7 does not prevent recurrence of CDI by inhibition of sporulation, but by sparing of members of the protective gut microbiota (vide infra).”*

2. Nature communications state that the title and/or abstract must indicate when findings apply to only one sex or gender. As this study was performed on Male mice, can the authors please add the gender of the mice to the abstract.

Author Response: *The data generated was indeed in a model using only male animals. This is for the reasons provided in our response to reviewer 1 (see above). The findings reported in our manuscript are, however, expected to apply equally to all sexes/genders. As per the reviewer's request, we have added 'in male mice' to the abstract.*

Protocol sporulation assay (broth dilution)

Sporulation assays of *C. difficile* were performed at the LUMC (Leiden, NL). For spore enumerations, laboratory *C. difficile* strain 630 Δ erm (van Eijk, 2015; Hussain, 2005) was recovered on CLO plates (bioMérieux). Subsequently, serially diluted overnight cultures were set up (5-fold dilutions) in glass tubes with 5 mL of BHI medium supplemented with 0.5% yeast extract (BHIY). After overnight growth, the lowest dilution with an approximate OD₆₀₀ of 1 was used to inoculate a large volume of fresh, prereduced BHIY medium. This ensured rapid and synchronous growth, in comparison with using a late stationary phase culture. The culture was subsequently split over 6 independent flasks, three of which were supplemented with **EVG7** at 0.125 \times MIC (0.016 mg/L). This concentration was experimentally determined to result in growth kinetics indistinguishable from a non-treated culture, with no significant difference in plating efficiency, which might otherwise have affected the sporulation kinetics. Cultures were grown for up to 72 h and viable cell counts and spore counts were performed at 24 h intervals. In short, for viable cell counts, cells were 10-fold serially diluted in a 96-well microtiter plate in prereduced BHIY and 100 μ L volumes of the 10⁻⁴ to 10⁻⁶ dilutions were plated on BHIY agar plates (1.5% agar) supplemented with 0.1% sodium taurocholate as a germinant. For spore counts, 200 μ L aliquots of the culture were heat treated for 20 minutes at 65 °C, after which they were serially diluted in prereduced BHIY. 100 μ L volumes of 10⁰ to 10⁻² (T = 24 h) or 10⁻¹ to 10⁻³ (T = 48 h and 72 h) were plated on BHIY agar plates supplemented with 0.1% sodium taurocholate as a germinant. Colonies were enumerated using a SCAN500 (Interscience) colony counter after 48–72h. Plates with countable colonies (i.e. non-confluent and clearly separated) colonies were used for calculations (generally 10⁻¹ for T24h spores, 10⁻³ for T48h or T72h spores; 10⁻⁵ or 10⁻⁶ for viable cell counts across all conditions).

Response to Reviewers Fig. 1: EVG7 does not inhibit *C. difficile* sporulation.

a, Vegetative cells and **b**, heat-resistant spores of *C. difficile* strain *C. difficile* strain 630 Δ erm^{56,57} in colony-forming units (CFUs) per mL selective BHIY media, in absence of **EVG7** (grey) or in presence of 0.125 \times MIC **EVG7** (blue). Bacterial enumeration was performed on selective media to measure vegetative cells and spores at 24 h, 48 h, and 72 h. Each dot represents an individual measurement (n = 3), bars represent the mean with error bars showing the standard deviation. Significance was determined using non-parametric two-tailed Mann-Whitney test (ns = not significant).